# A Fine-Scale Hotspot at the Edge: Epigean Arthropods from the Atacama Coast (Paposo-Taltal, Antofagasta Region, Chile)

**DOI:** 10.3390/insects12100916

**Published:** 2021-10-08

**Authors:** Jaime Pizarro-Araya, Fermín M. Alfaro, Andrés A. Ojanguren-Affilastro, Andrés Moreira-Muñoz

**Affiliations:** 1Laboratorio de Entomología Ecológica, Departamento de Biología, Facultad de Ciencias, Universidad de La Serena, Casilla 554, La Serena 1700000, Chile; fmalfaro@userena.cl; 2Instituto de Investigación Multidisciplinar en Ciencia y Tecnología, Universidad de La Serena, Casilla 554, La Serena 1700000, Chile; 3División de Aracnología, Museo Argentino de Ciencias Naturales “Bernardino Rivadavia” (MACN, CONICET), Avenida Ángel Gallardo 470, CABA, Buenos Aires 1405DJR, Argentina; andres.ojanguren@gmail.com; 4Instituto de Geografía, Pontificia Universidad Católica de Valparaíso, Avenida Brasil 2241, Valparaíso 2340000, Chile

**Keywords:** insect declines, homogenocene, hidden diversity, coastal desert, arthropod diversity, fog oases, lomas formation

## Abstract

**Simple Summary:**

The identification and surveying of fine-scale or micro-hotspots of biodiversity is a crucial strategy for better driving conservation efforts at global hotspots. This seems to be especially relevant at the edges of environments suitable for life, i.e., in desert margins with high levels of endemism, such as the Atacama coast. We surveyed a 100 km section of the Atacama coast including the emblematic Paposo and Taltal sites. We studied the taxonomic composition, richness, and abundance of terrestrial arthropods and were able to identify 173 arthropod species grouped into 118 genera and 57 families. The most abundant orders were Hymenoptera, Coleoptera, and Collembola, which accounted for 90.0% of the total captured. Most abundant families were Melyridae (Coleoptera), Poduridae (Collembola), Tenebrionidae (Coleoptera), and an indeterminate family of Psocoptera. Among remarkable Coleoptera, we were able to register *Ectinogonia barrigai* (Buprestidae) and *Luispenaia paposo* (Scarabaeidae). We also documented the presence of the tenebrionid *Gyriosomus angustus*, and several species of the genera *Nycterinus* (*Paranycterinus*) and *Scotobius*. We also could find the four species of scorpions that have been described for the Paposo area recently, and an undescribed species. The relevance of the area for future prospections and as a conservation site and a fine-scale hotspot of biodiversity has been confirmed based on the epigean arthropods.

**Abstract:**

The Atacama Desert at its margins harbors a unique biodiversity that is still very poorly known, especially in coastal fog oases spanning from Perú towards the Atacama coast. An outstanding species-rich fog oasis is the latitudinal fringe Paposo-Taltal, that is considered an iconic site of the Lomas formation. This contribution is the first to reveal the knowledge on arthropods of this emblematic coastal section. We used pitfall traps to study the taxonomic composition, richness, and abundance of terrestrial arthropods in 17 sample sites along a 100 km section of the coast between 24.5 and 25.5 southern latitude, in a variety of characteristic habitats. From a total of 9154 individuals, we were able to identify 173 arthropod species grouped into 118 genera and 57 families. The most diverse group were insects, with 146 species grouped in 97 genera and 43 families, while arachnids were represented by 27 species grouped into 21 genera and 14 families. Current conservation challenges on a global scale are driving the creation and evaluation of potential conservation sites in regions with few protected areas, such as the margins of the Atacama Desert. Better taxonomic, distributional, and population knowledge is urgently needed to perform concrete conservation actions in a biodiversity hotspot at a desert edge.

## 1. Introduction

Arthropods, the main component of global biodiversity, are declining in abundance and biodiversity in many regions of the world [1,2] and they are still a very unknown group. In this new geological epoch, called the *Anthropocene* [3], or *Homogenocene* [4,5], arthropods continue to be the “hidden biodiversity” not only in the tropics [6,7], but also in subtropical arid environments that have been traditionally overlooked by conservation efforts [8,9]. Similarly, there are still huge information and knowledge gaps regarding the biodiversity of arthropods in desert ecosystems (i.e., Linnean, Wallacean, and Prestonian shortfalls) [10,11].

Current conservation challenges on a global scale are driving the creation and evaluation of potential conservation sites in environments with few protected areas, such as the Atacama Desert. Recognized as the driest desert on the planet, the Atacama Desert has a biodiversity that is still poorly known, but emblematic due to its adaptation to the lack of humidity and its consequent high levels of endemism [12,13,14].

The Chilean Winter Rainfall and Valdivian Forests biodiversity hotspot is considered among the 25 global-scale conservation priority zones [15]. It encompasses a wide variety of environments, biotic communities, and species from the southern temperate forests (47° lat S), to the margin of the Atacama Desert along the coast of Antofagasta to the Mejillones Peninsula, at a latitude close to the Tropic of Capricorn (23° lat S) (Figure 1). The northern limit of this biodiversity hotspot is the margin of the Antofagasta coastal desert [15,16], which is also part of the Lomas Formation, an area extending from southern Peru to the north of La Serena city, Chile (El Tofo, 29.5° lat S) [17,18,19] (Figure 1). The Lomas Formation spans tropical and subtropical latitudes and extends into the most arid part of the Atacama Desert as a result of the stable position of the Pacific Anticyclone [19]. The formation has a conspicuous vegetation that is expressed in the so-called ‘fog oases’, which are dependent on the continuous presence of sea fogs, or *camanchacas* (Figure 1). These fog oases are sites located between sea level and 1000 m asl where vegetation and biotic richness is remarkably higher than on the surrounding harsh environment mainly due to a local topography (coastal cliffs) that favors the inland penetration and entrapment of these *camanchacas*.

These locally more humid conditions are exacerbated under the effects of the ENSO events (El Niño-Southern Oscillation) [18,20], which is also the main driver of the so-called blooming, or flowering desert, affecting plant as well as arthropod assemblages at the desert’s edge [21,22,23,24,25,26,27,28]. On the other hand, evidence also shows the effects of recent regional climate changes in these environments [29], including extreme temperature and precipitation events, such as those occurred in March 2015 and 2017, with consequent damaging streamflows [30,31].

One of the most relevant biotic sections along the Atacama coast is the Paposo-Taltal coastal site, located between 24.5° and 25.5° lat S. This site has been long recognized as a remarkable vegetational and floristic spot of the coastal margin of the Antofagasta Region [32,33,34,35] and has been recently highlighted as one of the most plant species-rich fog oases along 3000 km of the Peru-Chile coastal belt [36]. A remarkable arthropod fauna is associated to this floristic richness, including several endemic bee species, like *Liphanthus jenamro* Mir Sharifi, Graham and Packer [37] and *Neofidelia camanchaca* Dumesh and Packer [38]; two scarabs [39]; six tenebrionids [33,40,41,42,43,44]; four latridids [45]; a buprestid [46]; four spiders [47,48,49,50]; a monotypic genus of oribatid mite [51]; a freshwater amphipod [52]; three endemic scorpions [53,54].

This section of the coastal desert has been already recognized in biogeographic analyses as an outstanding area of endemism [33,55], but to date there is no systematic record of the entomofauna present in this coast, which is a major limitation for putting forward concrete conservation actions for the northern part of the Mediterranean Chilean hotspot. Although its botanical importance has been recognized for more than a century, a better understanding of the entire biota is essential for conducting more effective and concrete conservation actions. Although several studies, reports, and theses have reported on the arthropod richness and abundance in fog oases along the Peru coast [56,57,58,59,60], no systematic survey has been undertaken for the Chilean fog oases. This work aims to fill this gap by providing preliminary data about the epigean arthropod assemblage of the coastal desert at the Paposo-Taltal section (24.5°–25.5° latitude S) to help inform conservation efforts for the northern edge of the Mediterranean hotspot of biodiversity. 

## 2. Materials and Methods

### 2.1. Study Area

The Paposo-Taltal site is located around 200 km south of the city of Antofagasta, Chile (Figure 1). Ecologically, the area belongs to the formation of the Coastal Desert of Taltal, in the sub-region of the Coastal Desert and the Desert region [61].

The longitudinal coastal belt of the Atacama Desert is made up of distinct ecological, climatic, and geomorphological zones [62,63,64]. The range of habitats present in these zones and the relative stability of arid conditions since the Late Jurassic [63] have allowed the evolution of a particularly diverse biota adapted to the arid conditions and the fluctuations in humidity and dryness of this coastal desert [23]. This particular area is characterized by the presence of a series of fog oases that can be considered as fine-scale hotspots of diversity and endemism in a latitudinal gradient at the northern margin of a global biodiversity hotspot [64].

According to [65], the site vegetation is represented by the Desert Shrubland formation in the coastal plains and cliffs, and the Lower Desert Shrubland formation in the interior plains. The Desert Shrubland has three altitudinal levels of vegetation, namely, ‘Mediterranean coastal desert shrub of *Gypothamnium pinifolium* Phil. and *Heliotropium pycnophyllum* Phil.’, which extends from the sea level to 300 m asl; ‘Mediterranean coastal desert shrub of *Euphorbia lactiflua* Phil. and *Eulychnia iquiquensis* ((K. Schum.) Britton and Rose), which is heavily influenced by coastal fog and extends from 300 to 800 m asl, and ‘Mediterranean interior desert shrub of *Oxyphyllum ulicinum* Phil. and *Gymnophyton foliosum* Phil.’, which occupies the upper portion of the coastal cliffs. The Lower Desert Shrubland formation presents only the vegetation belt corresponding to ‘Matorral Bajo Desértico Tropical Interior de *Nolana leptophylla* (Miers) I.M. Johnst. and *Cistanthe salsoloides* (Barnéoud) Carolin ex Hershkovitz’ [65]. The climate in the Paposo-Taltal site corresponds to the coastal semidesert type with low thermal amplitude and abundant fogs resulting from the influence of the Humboldt Current [66] at the transition from the Mediterranean to the xeric subtropical bioclimates of northern Chile [65,67]. The presence of more abundant coastal fog than in surrounding areas, due to the particular orography of this site, provides considerable moisture for the development of the different plant formations [29,61]. At the weather station of Taltal, mean precipitation is 25 mm [65], but the annual and interannual variability is much higher, as is the moist content, dependent mainly on El Niño phenomenon [20,29]. 

### 2.2. Methods Used for Sampling Terrestrial Arthropods

The taxonomic composition (at species/family/order level) and relative abundance of the arthropod communities were determined from specimens captured using pitfall traps in 17 sample sites along ca. 100 km (Table 1, Figure 1D and Figure 2). The samplings were carried out in three campaigns: year 1 (2015) with 9 sites, year 2 (2017) with 4 sites, and year 3 (2019) with 4 sites (see Table 1). For years 1 and 3 in each of the sites, two 4 m × 5 m plots were installed, each consisting of 20 interception traps arranged in a grid, while for year 2, four interception traps were installed 50 m away from the sampling reference point at each site. Each trap consisted of two plastic cups one inside the other. The inner cup was filled up to two thirds with a mixture of water, household detergent and ethyl alcohol *sensu* [21,22]. The traps operated for four days (three active nights) during October 2015, August 2017, and December 2019. The collected material was processed, determined, and deposited in the entomological collection of the Ecological Entomology Laboratory of the University of La Serena (LEULS, Jaime Pizarro-Araya) and in the Arachnology Division of the “Bernardino Rivadavia” Argentine Museum of Natural Sciences, Buenos Aires, Argentina (MACN-Ar, Martín J. Ramírez).

### 2.3. Taxonomic Identification

The taxonomic determination was carried out using a stereomicroscope magnifying glass connected to a camera lucida. The taxa nomenclature follows the specific references listed as Appendix A in Appendix A. Some taxa (e.g., Araneae, Collembola, and Hymenoptera) were taxonomically identified only at the level of morphotype/family due to the poor taxonomic knowledge on these groups.

## 3. Results

### Taxonomic Composition and Richness

A total of 9146 individuals of arthropods were collected, representing 173 species grouped into 118 genera and 57 families. Insects were the group with the highest number of taxa (146 species, 97 genera, 43 families), while arachnids were represented by 27 species grouped into 21 genera and 14 families. Eleven orders were identified, of which Coleoptera and Hymenoptera were the most diversified, with 96 and 21 species, respectively. Among arachnids, the most represented orders were Araneae and Acari, with 11 and 7 species, respectively (Table 2, Appendix A Appendix A).

The insect assemblage was the most abundant arthropod group, with 99.2% of the total captured. The most abundant orders were Hymenoptera, Coleoptera, and Collembola, which accounted for 90.0% of the total captured (Table 3, Figure 3). The high number of individuals registered in Hymenoptera was due to Formicidae (37.8% of the total captured), represented by species such as *Brachymyrmex* sp., *Camponotus morosus* Smith, 1858, *Dorymyrmex goetschi* Goetsch, 1933, *Dorymyrmex pogonius* Snelling, 1975, and *Solenopsis gayi* Spinola, 1851. Although the eusocial characteristics of Formicidae (Hymenoptera) make comparisons between desert arthropod communities difficult [68,69], we have included them in the analysis due to the scarce information on the ecological aspects of arid zones [70] and for contributing new information to the current inventory. Other abundant families were Melyridae (Coleoptera), Poduridae (Collembola), Tenebrionidae (Coleoptera), and an indeterminate family of Psocoptera (Table 3).

The richness and relative abundance of arthropods varied between the different sites under study. The sites with the highest number of species were PPSIR5 (36 species), PPSCR (32 species) and PPSIR6 (31 species). Most of the sites presented over 20 species. In terms of relative abundance, PPSIR5 presented the highest value (1320 individuals), followed by PPSIR7 (1166) and PPSIR6 (1028) (Figure 4).

Within Coleoptera, the presence and restricted distribution of *Ectinogonia barrigai* Moore, 2017 (Buprestidae) and *Luispenaia paposo* Mondaca, Pizarro-Araya and Alfaro, 2019 (Scarabaeidae) was registered; these are recently discovered species occurring in fragmented sand-dune coastal environments of the northern portion of the section. We also documented the presence of the tenebrionids *Gyriosomus angustus* Philippi, 1864, *Nycterinus* (*Paranycterinus*) *angusticollis* Philippi, 1864, *Nycterinus* (*Paranycterinus*) *barriai* Peña, 1971, *Nycterinus* (*Paranycterinus*) *borealis* Peña, 1971, *Scotobius kaszabi* Marcuzzi, 1976, and *Scotobius tarapacensis* Marcuzzi, 1976, all of them species of restricted distribution in these environments.

Up to now four species of scorpions have been described for the Paposo area, namely, *Bothriurus dumayi* Cekalovic, 1974; *Brachistosternus barrigai* Ojanguren-Affilastro and Pizarro-Araya, 2014, and *Brachistosternus paposo* Ojanguren-Affilastro and Pizarro-Araya, 2014, all of them from Paposo [53,71], and *Brachistosternus philippii* Ojanguren-Affilastro, Pizarro-Araya and Ochoa, 2018, from Paposo Norte Natural Monument [54]. Additionally, *Caraboctonus keyserlingi* Pocock, 1893, a widespread species from Chile, has also been collected in the area [53], and recently we discovered an undescribed species of the genus *Rumikiru* Ojanguren-Affilastro, Ochoa, Mattoni and Prendini, 2012. This raises to six the number of scorpion species in the community of the area, one third of them being endemic. The presence of these recently discovered species shows that the scorpions in this area were very little explored. 

Some darkling beetles, like *Nycterinus* (*Paranycterinus*) *angusticollis*, *Nycterinus* (*Paranycterinus*) *barriai*, and *Scotobius kaszabi,* are species poorly represented in collections whose distributions are restricted to the coastal environments between Taltal and Paposo [41,72].

The remarkable diversity of scorpions in this area, compared to surrounding areas, seems to be related to the high variety of microhabitats that can be found in very close proximity, since most scorpion species are considered stenotopic. Additionally, this area represents the northernmost spot of distribution of several species that have higher humidity needs than the desert species of the area. One of these species, *C. keyserlingi*, extends far south into the Coquimbo and O’Higgins regions in central Chile, almost 1000 km to the south [73], and can only be found in the most humid spots of Paposo. Other species, such as *B. paposo* and *B. dumayi,* even reach the transitional coastal desert, located about 400 km to the south. 

Two of the described species of the area, *B. barrigai* and *B. philippii,* are presently considered endemic to Paposo; but their actual distribution and systematic position need to be further studied. A preliminary molecular analysis of *B. barrigai* place it as a probable synonym of northernmost specimens of *Brachistosternus kamanchaca* Ojanguren-Affilastro, Mattoni and Prendini, 2007, from areas nearby the Pan de Azúcar National Park [74]; however, further, more complete molecular studies are still necessary to elucidate its actual identity. Regarding *B. philippii*, the limits of its distribution range are still not completely known. Up until now, it has only been collected in a narrow fringe of coastal dunes in Paposo Norte Natural Monument [54]. This species has not been found in Paposo Norte in the coast near Antofagasta, and its range is probably restricted to a stripe of coast of less than 100 km. Based on this, we consider that, most probably, this is an endemic species of the area of Paposo Norte, but more field collections are still necessary to reveal its actual distribution. Finally, in a recent collection trip, we collected for the first time a species of the genus *Rumikiru* in the hills of the Paposo Norte Natural Monument (Figure 1D and Figure 5A,B). This species, still undescribed, is clearly different from *Rumikiru lourencoi* Ojanguren-Affilastro, 2003, found in the Pan de Azúcar National Park, and also different from another new congeneric species recently collected by our group in La Chimba National Reserve. Therefore, we consider that, most probably, this is also an endemic species of this particular area.

## 4. Discussion

The Paposo-Taltal section in the coast of Antofagasta, at the Atacama margin, is considered as one of the most outstanding plant species-rich fog oasis along 3000 km of the Perú-Chile coast [36]. The arthropod diversity reported here (173 species grouped into 118 genera and 57 families) is proportionally higher than other surveyed fog oases. Only as a reference, reports from Quebrada Las Brujas (Peru) gave 60 genera and 51 families [58]. 

Directed samplings in these coastal desert environments have made it possible, on the one hand, to update the historical records of several lesser-known species and, on the other, to identify taxa with conservation problems in environments under multiple threats. Most of the surveyed sites (13 sites) were located in representative environments within the boundaries of the Paposo Priority Site, declared as a site of high interest for conservation but not yet receiving formal recognition nor any kind of effective protection. The northern portion of the area encompasses the Paposo Norte Natural Monument, which still lacks adequate signaling and ranger services (Figure 1D). The restrictions and challenges for preserving these environments and the rich biotic assemblage are evident.

Paposo has been declared a priority site for conservation for its high biodiversity of plant species. However, it lacks legal recognition as a protected wild area of the State of Chile, even though various threats to this fragile environment have been reported [75]. Assessing the degree of completeness of local richness inventories is a key task to effectively direct sampling efforts in protected natural areas.

The high abundance recorded in the studied sites was represented by species such as *Arthrobrachus limbatus* Solier, 1849, a melirid with a wide distribution in northern and central Chile [42,76], native ants of the genus *Dorymyrmex* (Formicidae), and lesser-known collembolans in the families Poduridae, Entomobryidae, and Sminthuridae. The presence of most of these species could be a result of the floristic characteristics of these environments, which host a shrubby thicket with a predominance of herbaceous and shrubby plant species. Thus, the high abundance of edaphic species, such as collembolans, could be due to their preference for humid environments [77,78], however, it is also possible that the seasonal humidity or the water supply from the coastal fogs observed in these sites [20] (Figure 1D, Figure 2) could play an important role in the biological cycles of these species.

The Paposo area is home to several species of arthropods in category of conservation: *Gyriosomus angustus* Philippi, 1864 (Coleoptera: Tenebrionidae) (EN = Endangered); *Scotobius planicosta* Guérin-Méneville, 1834 (Coleoptera: Tenebrionidae) (EN = Endangered); *Conometopus penai* Ronderos, 1972 (Orthoptera: Ommexechidae) (EX = Extinct); *Brachistosternus philippii* Ojanguren-Affilastro, Pizarro-Araya and Ochoa, 2018 (Scorpiones: Bothriuridae); *Ectinogonia barrigai* Moore, 2017 (Coleoptera: Buprestidae); *Enodisomacris curtipennis* Cigliano, 1989 (Orthoptera: Tristiridae); *Luispenaia paposo* Mondaca Pizarro-Araya and Alfaro, 2019 (Coleoptera: Scarabaeidae), and *Nycterinus* (*Paranycterinus*) *angusticollis* Philippi and Philippi, 1864 (Coleoptera: Tenebrionidae) (see Figure 5). These species have been categorized by the Chilean Ministry of Environment due to the multiple threats on their habitats [79] implicitly accepting the endangered situation of the biota of this area.

The identification and surveying of fine-scale or micro-hotspots of biodiversity is a crucial goal for better driving conservation efforts at global hotspots [80,81,82,83,84,85]. This seems to be especially relevant at the edges of the global hotspots and at the edge of habitats suitable for life, i.e., in desert margins with high levels of endemism, such as the Atacama coast. Nevertheless, for better conservation planning, several shortfalls still need to be overcome in many regions (i.e., Linnean, Wallacean, and Prestonian shortfalls). Long-term ecological and biogeographic studies for individual species and populations are desirable [86].

Lastly, the relationship between arthropod diversity and endangered host plants has been explored in other Mediterranean fine-scale hotspots [87], and several reports on this topic are available for Atacama [88]. However, much more field work on the coast is needed for disentangling those relationships, especially in El Niño years. The potential effects of climate change are already under observation in different coastal regions of Latin America [89] and hotspots worldwide [90,91], but much needs to be done especially in the field, considering that the changes in plant species composition seem to drastically affect endangered insects [92]. All these are crucial aspects that will certainly drive entomological surveying and monitoring in our current times of ecological crises.

## Figures and Tables

**Figure 1 insects-12-00916-f001:**
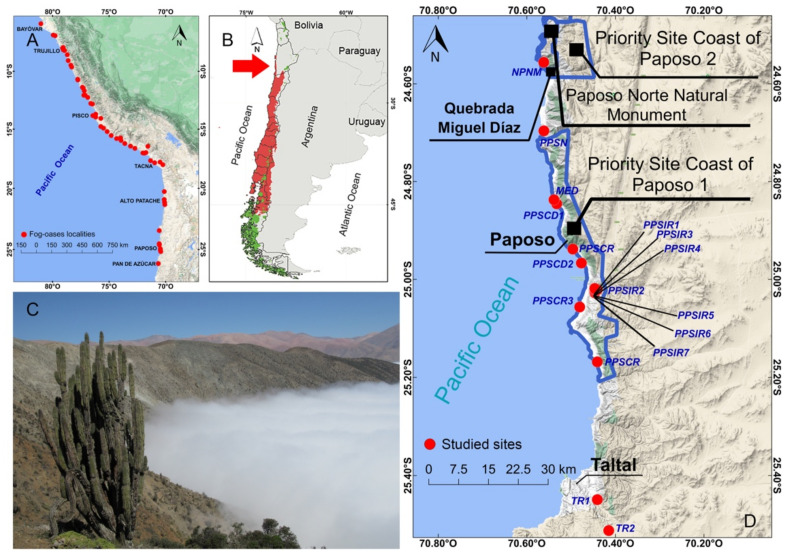
Regional context of the study area: (**A**) fog-oases along the Peruvian and Chilean arid coast (red dots). (**B**) The Mediterranean Chilean biodiversity hotspot (marked in red); Paposo-Taltal section highlighted by red arrow. (**C**) Camanchaca over the coastal cliffs of the study site. (**D**) Location of the studied sites in the coastal section North of Paposo to South of Taltal, and limits of Paposo Norte Natural Monument and Priority Site Coast of Paposo, 1 and 2 (Antofagasta Region, Chile).

**Figure 2 insects-12-00916-f002:**
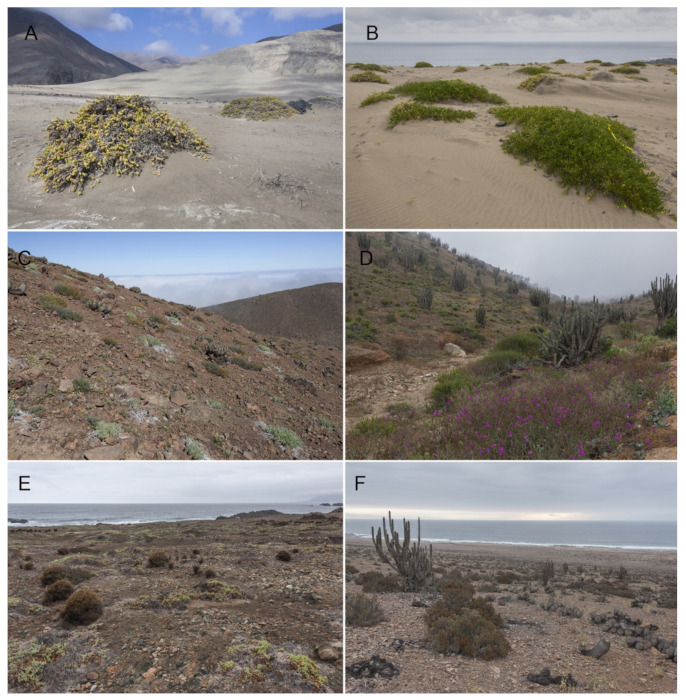
Study sites within Paposo in the Antofagasta Region, Chile: (**A**) Paposo Norte Natural Monument (NPNM) during 2015. (**B**) Paposo Priority Site (coastal dune) (PPSCD1) during 2015. (**C**) Paposo Priority Site (interior ravine) (PPSIR2) during 2017. (**D**) Paposo Priority Site (interior ravine) (PPSIR4) during 2015. (**E**) Paposo Priority Site (coastal dune) (PPSCD3) during 2019. (**F**) Paposo Priority Site (Cachinales ravine) (PPSCR) during 2019.

**Figure 3 insects-12-00916-f003:**
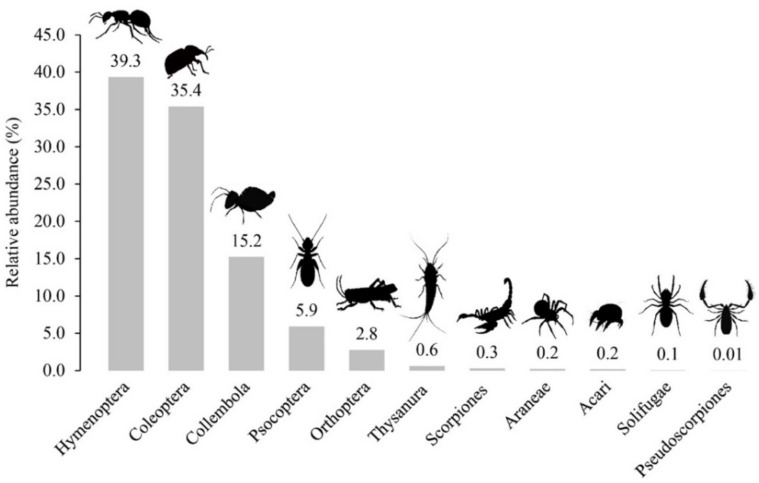
Relative abundance (individuals) of epigean arthropod orders registered at the Paposo-Taltal section (Antofagasta Region, Chile).

**Figure 4 insects-12-00916-f004:**
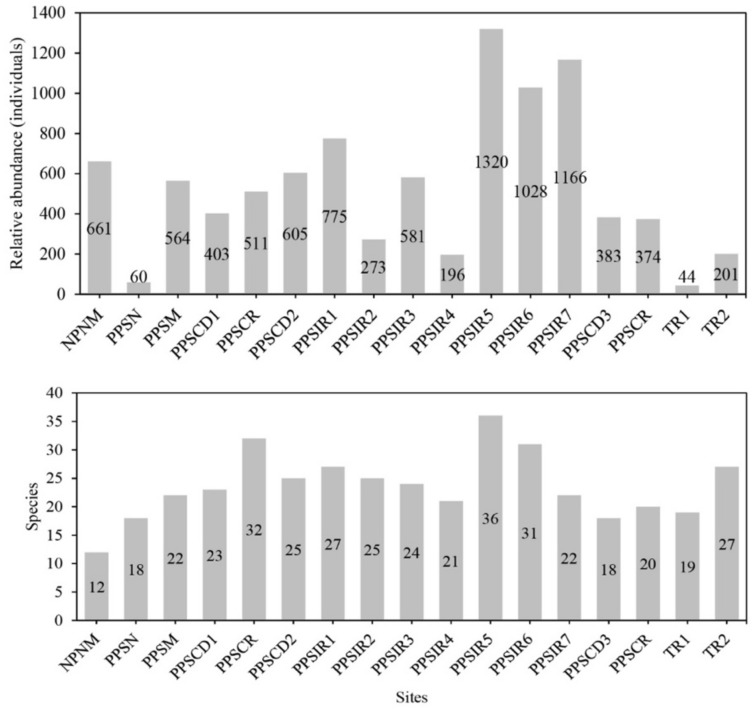
Relative abundance (individuals) and number of epigean arthropod species for each study site at Paposo-Taltal (Antofagasta Region, Chile). Codes as in Table 1.

**Figure 5 insects-12-00916-f005:**
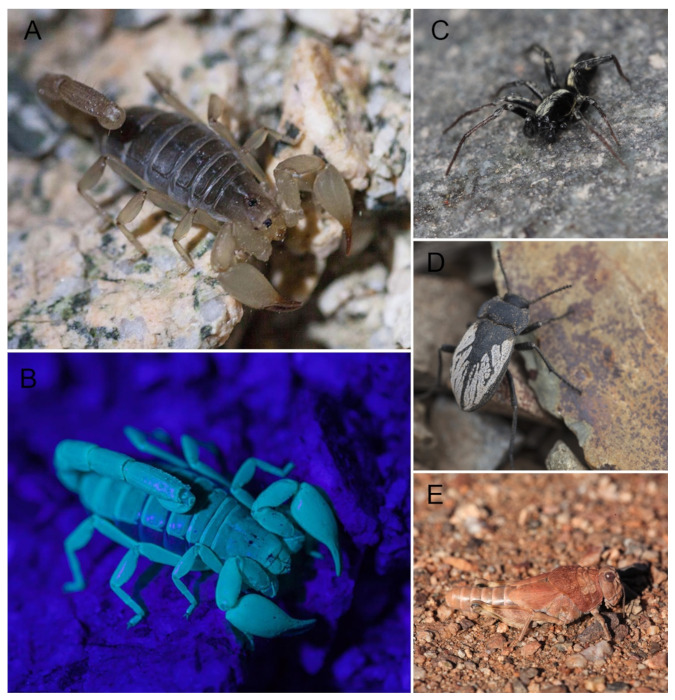
Different species of arthropods recorded in Paposo (Antofagasta Region, Chile). (**A**) Female of *Rumikiru* sp. nov. (Scorpiones: Bothriuridae) endemic of Paposo Norte Natural Monument under white light and (**B**) under UV light. (**C**) Male of *Aysenia paposo* Laborda, Ramírez and Pizarro-Araya (2013) (Araneae: Anyphaenidae). (**D**) Male of *Gyriosomus angustus* Philippi, 1864 (Coleoptera: Tenebrionidae). (**E**) Female of *Enodisomacris curtipennis* Cigliano, 1989 (Orthoptera: Tristiridae).

**Table 1 insects-12-00916-t001:** Habitat description and geographical location for the studied sites in Paposo (Antofagasta Region, Chile).

N° Site			Site Code		
1	NPNM	Latitude	Longitude	Year	Habitat Description
2	PPSN	−24.578962°	−70.552316°	2019	Sandy beach with open scrub of *Tetragonia* sp. (Aizoceae).
3	PPSM	−24.694903°	−70.561567°	2017	Coastal plain (40 m asl) with scarce vegetation.
4	PPSCD1	−24.840082°	−70.536391°	2019	Cliffs (70 m asl) with sandy-rocky substrate with scarce vegetation.
5	PPSCR	−24.851444°	−70.526972°	2015	Coastal plain, (50 m asl), sandy substrate, open scrub of *Skytanthus acutus* (Apocynaceae).
6	PPSCD2	−24.939056°	−70.490389°	2015	Coastal plain, alluvial cone (140 m asl), open scrub of *Euphorbia lactiflua* (Euphorbiaceae).
7	PPSIR1	−24.968222°	−70.476667°	2015	Coastal plain with reefs (escollos) (15 m asl), low scrub of *Cristaria integerrima*, *Tetragonia* sp., and *Nolana* sp. (Solanaceae).
8	PPSIR2	−25.002861°	−70.446944°	2015	Ravine to leeward (676 m asl), low scrub of *Frankenia chilensis* (Frankeniaceae) and *Mesembryanthemum crystallinum* (Aizoceae).
9	PPSIR3	−25.006307°	−70.426788°	2017	Interior ravine (1007 m asl), open scrub of *Copiapoa* sp. (Cactaceae) and *Heliotropium* sp. (Heliotropiaceae).
10	PPSIR4	−25.005639°	−70.447556°	2015	Ravine to leeward (664 m asl), open scrub of *Proustia cuneifolia* ssp. *tipia* (Asteraceae) and cacti (*Eulychnia iquiquensis*).
11	PPSIR5	−25.006528°	−70.446583°	2015	Ravine to leeward (645 m asl), interior scrub of *Ophryosporus triangularis* (Asteraceae) with cacti (*Eulychnia iquiquensis*) and *Cistanthe* sp. (Montiaceae).
12	PPSIR6	−25.010556°	−70.447583°	2015	Ravine to leeward (592 m asl), interior scrub of *Euphorbia lactiflua* with cacti (*Eulychnia iquiquensis*).
13	PPSIR7	−25.010556°	−70.447194°	2015	Ravine to leeward (588 m asl), interior scrub of *Ophryosporus triangularis* with cacti (*Eulychnia iquiquensis*).
14	PPSCD3	−25.010417°	−70.446111°	2015	Ravine to leeward (595 m asl), interior scrub of *Euphorbia lactiflua* with cacti (*Eulychnia iquiquensis*).
15	PPSCR	−25.055935°	−70.481731°	2019	Coastal terrace (8 m asl) with low open scrub of *Nolana crassulifolia* and *Bakerolimon plumosum* (Plumbaginaceae).
16	TR1	−25.171344°	−70.436829°	2019	Coastal plain (88 m asl) with open scrub of *Heliotropium* sp., *Eulychnia* sp. and *Copiapoa* sp.
17	TR2	−25.448125°	−70.435046°	2017	Interior ravine (620 m asl) with open scrub of *Euphorbia lactiflua*, *Polyachyrus* sp. (Asteraceae), *Nolana crassulifolia*, and *Eulychnia iquiquensis.*

**Table 2 insects-12-00916-t002:** Number of taxa (family, genus, and species) and ratios between the taxonomic levels of epigean arthropods from Paposo (Antofagasta Region, Chile).

Taxa					
Class-Order	Family	Genus	Species	Genus/Family	Species/Family
Arachnida	**14**	**21**	**27**	**1.50**	**1.92**
Acari	2	3	7	1.50	3.50
Araneae	7	11	11	1.83	1.83
Pseudoscorpiones	1	1	1	1.00	1.00
Scorpiones	2	4	6	2.00	3.00
Solifugae	2	2	2	1.00	1.00
Insecta	**43**	**97**	**146**	**2.25**	**3.39**
Coleoptera	26	62	96	2.38	3.69
Collembola	5	8	8	1.60	1.60
Hymenoptera	3	11	21	3.67	7.00
Orthoptera	6	10	13	1.67	2.17
Psocoptera	2	4	6	2.00	3.00
Thysanura	1	2	2	2.00	2.00

Note: bold—subtotals.

**Table 3 insects-12-00916-t003:** Richness of species and relative abundance (individuals captured) for the different families of epigean arthropods registered in Paposo (Antofagasta Region, Chile).

Order	Family	Species	Relative Abundance	Percentage
Acari	Caeculidae	1	2	0.02
	Indeterminate	6	14	0.15
Araneae	Araneidae	3	6	0.07
	Thomisidae	1	1	0.01
	Gnaphosidae	2	5	0.05
	Sicariidae	2	3	0.03
	Theridiidae	1	3	0.03
	Anyphaenidae	1	1	0.01
	Indeterminate	1	1	0.01
Pseudoscorpiones	Cheiridiidae	1	1	0.01
Scorpiones	Bothriuridae	5	23	0.24
	Caraboctonidae	1	5	0.05
Solifugae	Daesiidae	1	1	0.01
	Mummuciidae	1	4	0.04
Coleoptera	Anthicidae	1	35	0.38
	Buprestidae	2	53	0.58
	Carabidae	4	283	3.09
	Cerambycidae	1	1	0.01
	Chrysomelidae	3	11	0.12
	Cleridae	1	7	0.08
	Corylophidae	1	3	0.03
	Cryptophagidae	2	4	0.04
	Curculionidae	26	205	2.24
	Elateridae	1	1	0.01
	Histeridae	1	5	0.05
	Latridiidae	5	25	0.27
	Leiodidae	1	101	1.10
	Mauroniscidae	1	5	0.05
	Meloidae	3	16	0.17
	Melyridae	3	1.800	19.68
	Mordellidae	1	5	0.05
	Nitidulidae	2	10	0.11
	Oedemeridae	1	26	0.28
	Phengodidae	1	27	0.30
	Ptiliidae	4	6	0.07
	Ptinidae	3	4	0.04
	Scarabaeidae	2	2	0.02
	Staphylinidae	3	62	0.68
	Tenebrionidae	22	537	5.87
	Zopheridae	1	2	0.02
Collembola	Entomobryidae	2	510	5.58
	Hypogastruridae	1	9	0.10
	Poduridae	2	591	6.46
	Sminthuridae	2	205	2.24
	Indeterminate	1	78	0.85
Hymenoptera	Bradynobaenidae	2	4	0.04
	Formicidae	13	3.446	37.68
	Mutillidae	6	147	1.61
Orthoptera	Acrididae	1	1	0.01
	Mogoplistidae	2	188	2.06
	Ommexechidae	2	9	0.10
	Proscopiidae	1	1	0.01
	Tettigoniidae	5	35	0.38
	Tristiridae	2	20	0.22
Psocoptera	Liposcelidae	2	5	0.05
	Indeterminate	4	535	5.85
Thysanura	Lepismatidae	2	56	0.61
Totals	57	173	9.146	100

## Data Availability

Collected specimens have been deposited in the entomological collection of the Ecological Entomology Laboratory of the University of La Serena (LEULS) and in the Arachnology Division of the “Bernardino Rivadavia” Argentine Museum of Natural Sciences, Buenos Aires, Argentina (MACN-Ar, Martín J. Ramírez). All published data are available upon formal request.

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
