# Peer review of "A Fine-Scale Hotspot at the Edge: Epigean Arthropods from the Atacama Coast (Paposo-Taltal, Antofagasta Region, Chile)"

_insects, 2021, doi:10.3390/insects12100916_

Round 1
Reviewer 1 Report
The file with some comments is attached

Author Response
Submission Date
11 August 2021
Date of this review
12 Aug 2021 09:58:02
Line 87: it is not a new species, it was described in 2019
Answer: Thank you for the correction. We have updated the scientific name of the species.
Line 88: in the previous species was used &, not and
Answer: Thank you for the correction. We have changed “and” by “&”.
Line 140: explain better what sensu means in this context
Answer: This Latin term is used to indicate "in the sense of"
Line 166: of Arthropods
Answer: We have corrected the sentence.
Line 210: The presence of these recently discovered species shows that the scorpions in this area were very little explored
Answer: We agree with the reviewer and we have included this sentence in the text.
Line 241: presently considered
Answer: We have changed the sentence (presently considered endemic).
Line 246: is the morphology identical? Generally, genetics are used when very little differences are found
Answer: Not the morphology is not identical but it is very similar, and they can only be separated by subtle morphological differences. We are currently studying if these morphological differences used to describe and separate both species were actually interpopulational differences, and not interspecific differences.
Line 307: the Chilean area is Mediterranean-like
Answer: We agree with the reviewer.
Line 344: many words that should be written in italics are highlighted
Answer: We have corrected the words.
Reviewer 2 Report
In the manuscript at hand the authors report on arthropod taxa from 17 sites along the pacific coast of Chile. The authors claim that this surveying is a crucial strategy for better driving conservation efforts in other biodiversity hotspots . However, ecological relationships or conservation managements are not evaluated in this study. Thus, it remains a species inventory of local interest and might be suitable for publication in a regional faunistic journal.
Author Response
Comments and Suggestions for Authors
In the manuscript at hand the authors report on arthropod taxa from 17 sites along the pacific coast of Chile. The authors claim that this surveying is a crucial strategy for better driving conservation efforts in other biodiversity hotspots . However, ecological relationships or conservation managements are not evaluated in this study. Thus, it remains a species inventory of local interest and might be suitable for publication in a regional faunistic journal.
Submission Date
11 August 2021
Date of this review
12 Aug 2021 15:00:54
Answer: We welcome your suggestions and insights on our manuscript. This work aims to provide unpublished data on the richness and abundance of epigeal arthropods in the coastal stretch between Taltal and Paposo (coastal desert of Chile), we also present the conservation efforts to date in the group. Said conservation efforts have focused mainly on the description and findings of new species (see lines 104-202) and on the categorization of species with conservation problems (see lines 289-299) based on criteria similar to those proposed by the IUCN (distribution, area of ​​occupation threatened environments), and carried out through the Ministry of the Environment of Chile. Our data and proposed objectives do not allow us to conduct a study aimed at evaluating conservation efforts since Paposo (and the coastal strip between Paposo and Taltal) does not currently have a legal figure in Chile that protects these environments. We consider that this work is not only a local inventory, but the first study to diagnose the current threats and particularities of the epigeal arthropod biota in this vulnerable coastal desert ecosystem.
Reviewer 3 Report
This is a basic but thorough study of an understudied habitat. It provides a solid foundation for further studies in ecology, conservation, and general arthropod biology.
This is a basic descriptive study of a habitat. It does not test any hypotheses, nor does it purport to deal with broad theory. It is thorough and enlightening. It provides a solid history and background for subsequent studies that have specific hypotheses to test and expand. Other than more years of study, I cannot suggest how its data could be better organized and presented.
Author Response
Comments and Suggestions for Authors
This is a basic but thorough study of an understudied habitat. It provides a solid foundation for further studies in ecology, conservation, and general arthropod biology.
This is a basic descriptive study of a habitat. It does not test any hypotheses, nor does it purport to deal with broad theory. It is thorough and enlightening. It provides a solid history and background for subsequent studies that have specific hypotheses to test and expand. Other than more years of study, I cannot suggest how its data could be better organized and presented.
Submission Date
11 August 2021
Date of this review
11 Aug 2021 21:15:49
Answer: Thank you very much for your observations and comments on our article. Indeed, the sampling effort to date is representing two years of sampling (2015 and 2019), efforts that in practice have allowed us, on the one hand, to carry out a general inventory for the first time of the scarcely studied arthropod communities, and on the other hand the discovery of several new species for the area.
Round 2
Reviewer 2 Report
I appreciate the effort by the authors to sample and identify multiple arthropod groups in a region with very limited knowledge about these groups. Pitfall traps are certainly a good method to sample ground-living arthropods. However, sites were sampled in different years, during different months and even with a slightly different sampling set-up in the second year. Such a ‘convenient sampling’ makes it difficult to compare sites among each other (L. 190-194) and complicates any statistical analysis. Site codes should appear in the map Fig. 1d and sampling year should be indicated for each site in Table 1. The result sections contains discussions of some discovered species (e.g. L. 221-239). The result section should only contain a report of the findings and the interpretation should be restricted to the discussion section. The discussion, on the other hand, contains descriptions of the study region, which are either redundant to the introduction or would be better situated in the method section (e.g. L. 277-284).
Unfortunately, I have to stay with my initial evaluation that this study is a local species inventory. Although the habitats are special and we lack information about the occurrence of these species in the region, the results remain descriptive and the interpretation of the findings is speculative. The presented data does not help to “diagnose the current threats” [response by the authors to my previous comments] for the sampled arthropods. A scientific valid evaluation of threats to these organisms would require population trends of the species of interest and/or a comparison of arthropod communities among sites that vary in the impact of specific threats – such an effort would require an adequate study design and a proper statistical analysis.
As it is, the manuscript is of local interest and lacks any scientific generalization or application to other regions, which would be of interest for a broader audience. I leave it to the editor to decide whether faunistic reports fit in the scope and the scientific standards of ‘insects’.
Author Response
We have made an effort to modify part of the results and discussion section and we have modified the position of the paragraphs in the text indicated by the reviewer 2. We agree with the reviewer 2 that the present study has different samples for each year, however, the objective of this work does not seek to evaluate differences between the studied sites from a statistical approach. We have modified the map with the codes of the studied sites, and we have also included the years of sampling in Table 1.
